# Safety and Efficacy of DOACs in Patients with Advanced and End-Stage Renal Disease

**DOI:** 10.3390/ijerph19031436

**Published:** 2022-01-27

**Authors:** Sylwester Rogula, Aleksandra Gąsecka, Tomasz Mazurek, Eliano Pio Navarese, Łukasz Szarpak, Krzysztof J. Filipiak

**Affiliations:** 1Department of Cardiology, Medical University of Warsaw, 02-097 Warsaw, Poland; sylwester.rogula@wum.edu.pl (S.R.); tomasz.mazurek@wum.edu.pl (T.M.); 2Department of Medicine, University of Alberta, Edmonton, AB T6G 2R7, Canada; eliano.navarese@cm.umk.pl; 3Henry JN Taub Department of Emergency Medicine, Baylor College of Medicine, Houston, TX 77030, USA; lukasz.szarpak@bcm.edu; 4Department of Clinical Sciences, Maria Sklodowska-Curie Medical Academy, 03-411 Warsaw, Poland; krzysztof.filipiak@uczelniamedyczna.com.pl

**Keywords:** direct oral anticoagulants, chronic kidney disease, end-stage renal disease, anticoagulation, hemodialysis, DOAC, CKD, ESRD

## Abstract

The prevalence of chronic kidney disease (CKD) is increasing due to the aging of the population and multiplication of risk factors, such as hypertension, arteriosclerosis and obesity. Impaired renal function increases both the risk of bleeding and thrombosis. There are two groups of orally administered drugs to prevent thromboembolic events in patients with CKD who require anticoagulation: vitamin K antagonists (VKAs) and direct oral anticoagulants (DOACs). Although VKAs remain the first-line treatment in patients with advanced CKD, treatment with VKAs is challenging due to difficulties in maintaining the appropriate anticoagulation level, tendency to accelerate vascular calcification and faster progression of CKD in patients treated with VKAs. On the other hand, the pleiotropic effect of DOACs, including vascular protection and anti-inflammatory properties along with comparable efficacy and safety of treatment with DOACs, compared to VKAs observed in preliminary reports encourages the use of DOACs in patients with CKD. This review summarizes the available data on the efficacy and safety of DOACs in patients with CKD and provides recommendations regarding the choice of the optimal drug and dosage depending on the CKD stage.

## 1. Introduction

Chronic kidney disease (CKD) is defined as kidney structure or function abnormality present for more than three months with health implications. Based on the estimated glomerular filtration rate (eGFR), CKD is classified into stage 1 (≥90 mL/min), stage 2 (60 to 89 mL/min), stage 3a (45 to 59 mL/min), stage 3b (30 to 44 mL/min), stage 4 (15 to 29 mL/min) and stage 5 (<15 mL/min) [1]. At stage 5 (end stage renal disease; ESRD), most patients require kidney replacement therapy, such as hemodialysis (HD), peritoneal dialysis or kidney transplantation.

The prevalence of CKD in adult Polish citizens was estimated at 5.8% in the NATPOL 2011 study, using the Chronic Kidney Disease Collaboration formula (CKD-EPI). This prevalence is increasing, partially due to the aging of society. After 40 years of age, glomerular filtration rate (GFR) decreases about 1 mL/1.73 m^2^/year, and after 65 years of age, even faster [2]. The main risk factors of CKD are atherosclerosis, hypertension, smoking, obesity and diabetes/impaired glucose tolerance [3,4,5,6,7]. The same risk factors contribute to the development of cardiovascular disease (CVD). Moreover, CKD is an independent risk factor of CVD. Hence, CVD and CKD commonly coexist.

Atrial fibrillation (AF) is present in 15–20% of patients with CKD [8,9]. AF may lead to the formation of thrombi in the left atrium and subsequently to systemic and cerebral embolization, including strokes. Strokes in patients with AF are more severe than strokes of another pathogenesis, such as atherosclerotic rupture or small vessel disease. In the US cohort study including 1061 patients, a 2.2-fold increase in confinement to bed following stroke was shown in patients with AF compared to patients with different stroke etiology [10]. In an Italian cohort study of 3530 patients, the mortality rate at one year was 49.5% in AF patients versus 27.1% in patients with strokes of different etiologies. Since CKD increases both the risk of thromboembolism and bleeding, the treatment of a patient with AF and CKD is even more challenging [11].

There is evidence of hemostasis disorders in kidney disease, supported by the observation of both in acute kidney injury and in CKD. Patients with acute kidney injury are, on one hand, prone to prolonged bleeding [12], and on the other hand, experience thrombosis [13]. Large epidemiologic studies showed that patients with ESRD or those requiring kidney replacement therapy have a ~2 times higher risk of stroke or systemic thromboembolism, compared to those with no kidney disease [14,15]. ESRD patients have also a ~3 times higher risk of bleeding, compared to those with no kidney disease [15]. Hemostasis disorders in patients with CKD result from the systemic accumulation of the uremic toxins and metabolic compounds, which cause activation of the coagulation cascade and fibrinolytic system, platelet hyperreactivity and damage of the endothelium [16]. Furthermore, renal failure is associated with chronic inflammation, which aggravates the vicious circle of dysfunctional hemostasis [17]. Finally, some patient might at the same time develop bleeding and thromboembolic episodes. Hemostasis disorders in CKD are shown in Figure 1.

For now, it is hard to predict which patient will develop bleeding problems and which will experience thromboembolism. Nevertheless, patients with CKD and AF require anticoagulation to prevent thromboembolic episodes. According to the guidelines, when oral anticoagulation is initiated in a patient with AF who is eligible for a direct oral anticoagulant (DOACs: apixaban, dabigatran, edoxaban or rivaroxaban), a DOAC is recommended in preference to a vitamin K antagonist (VKA) [18]. At present, there are no recommendations regarding anticoagulation in patients with CKD and AF. DOACs are preferred in CKD stages 1 to 4, with the exception for dabigatran, which cannot be used in stage 4 CKD. On the contrary, in patients with ESRD, VKAs remain the first-line treatment, based on expert consensus. However, it is unclear whether patients with CKD benefit from oral anticoagulation as much as those with normal kidney function. There are also limited data regarding the choice of the anticoagulants in patients with advanced CKD. This review outlines the benefit–risk ratio of anticoagulants in advanced CKD and provides practical recommendations for treatment adjustment, reversal of antithrombotic effect and monitoring of the renal function on a regular basis.

## 2. VKA in Patients with CKD

There are two main indications for VKA in which DOACs should not be used, including (i) valvular AF and (ii) patients with mechanical prosthetic valves. For example, patients with rheumatic mitral valve disease have the highest risk of venous thromboembolism (VTE) among those with any form of valvular heart disease, and the efficacy of DOACs has not been directly evaluated in patients with mitral stenosis. In patients with mechanical heart valves, in turn, DOACs increased the risk of thromboembolic and bleeding complications compared to warfarin [19]. In contrast to valvular AF and mechanical valves, advanced CKD is a relative contraindication to DOACs. Since patients with CKD were underrepresented in clinical trials comparing DOACs and VKAs, VKAs are traditionally used in patients with advanced CKD, based on the expert consensus. However, treatment of patients with CKD with VKAs has at least three disadvantages.

First, it is difficult to determine the VKA dosage to maintain the therapeutic range (TTR) in patients with CKD. The inappropriate level of anticoagulation results in suboptimal anticoagulation and increases both the risk of thromboembolic and bleeding complications [20].

Second, VKAs inhibit the activation of vitamin K-dependent calcification inhibitors. One of these inhibitors is matrix Gla protein (MGP), which is a well-established inhibitor of vascular calcification [21,22]. Inhibited activation of MGP in the presence of VKAs results in accelerated vascular calcification and induces a vulnerable plaque phenotype [23]. The most severe complication of vascular calcification is calciphylaxis (calcific uremic arteriopathy). Calciphylaxis is a condition characterized by necrosis of the skin and fatty tissue, affecting about 1% of patients treated with dialysis annually. The mortality due to calciphylaxis is up to 80%, often within several months of onset. Although the pathomechanism of calciphylaxis is not entirely understood, vitamin K deficiency, VKA use and MGP dysfunction are all associated with the disease [24]. Hence, it seems that the benefits of treatment with VKAs might not outweigh the risk of the progression of vascular calcification and potential complications.

Third, fibroblast growth factor-23 (FGF-23) concentrations increase markedly in chronic kidney disease [25]. FGF-23 is a hormone that promotes urinary phosphate excretion and regulates vitamin D metabolism. Higher levels of FGF-23 are associated with increased risk of clinical cardiovascular events [26]. FGF-23 can increase the incidence of atrial fibrillation (AF) by inducing left ventricular hypertrophy and diastolic and left atrial dysfunction. Furthermore, FGF-23 has been associated with endothelial dysfunction and vascular calcification, which, in cooperation with vascular calcification promoted by VKA use, can lead to faster progression of CKD and an increase of AF incidence.

Finally, faster progression of CKD was observed in patients using VKAs, who exceeded the target therapeutic window of international normalized ratio (INR of 2.0–3.0). The observation may be explained by recurring subclinical bleedings into the renal tubule system and subsequent tubular necrosis [27]. Furthermore, warfarin can cause acute kidney injury in patients with INR > 3.0 due to glomerular bleedings [28]. As a result, renal tubules are occluded. Serum creatinine increases, and acute kidney injury occurs. This process, known as warfarin-related nefropathy, leads to accelerated progression of CKD.

The disadvantages of VKA treatment in patients with advanced CKD are presented in Figure 2. Altogether, VKAs might not be the optimal treatment strategy in CKD.

## 3. Direct Oral Anticoagulants

The alternative to VKAs are DOACs. This group of drugs was brought to market in 2008 with dabigatran and rivaroxaban for the prevention of VTE after elective hip or knee replacement surgery. The direct thrombin inhibitor dabigatran and three inhibitors of factor Xa: rivaroxaban, apixaban and edoxaban are currently approved for use in the prevention of stroke and systemic embolism in adult patients with non-valvular AF [18]. The introduction of DOACs eliminated many difficulties associated with the use of VKAs. DOACs have a better pharmacokinetic profile due to the quick onset of action and therefore do not require bridge therapy with heparin in the first few days of treatment [29]. The inter-individual variability of clinical response to DOACs is lower as compared to VKA, resulting in a wider therapeutic window. DOACs also have fewer interactions with other drugs and food, providing a predictable therapeutic effect and allowing the constant dosing of the drug without INR monitoring or dietary restrictions [29]. However, because patients with advanced CKD were underrepresented in the hallmark clinical trials of DOACs, there are limited data about their efficacy and safety in patients with advanced CKD. According to the guidelines, creatinine clearance of <30 mL/min for dabigatran, rivaroxaban and edoxaban, and serum creatinine >2.5 mg/dL or creatinine clearance of <25 mL/min for apixaban is a contraindication to DOACs [30]. Although off-label, DOACs may potentially be useful in the treatment of patients with advanced CKD due to their anti-inflammatory activity, vascular protection and higher efficacy and safety, compared to VKA.

### 3.1. Anti-Inflammatory Activity of DOACs

The anti-inflammatory activity of DOACs was demonstrated in multiple studies. For example, dabigatran and rivaroxaban were shown to decrease the plasma concentration of pro-inflammatory markers, including adhesion molecules (ICAM-1, VCAM-1), cytokines (interleukin [IL]-8), chemoattractant chemokines (CCL2, CXCL2) and tissue factor [31]. Further, rivaroxaban reduced the expression of messenger RNA coding the pro-inflammatory mediators: tumor necrosis factor α and IL-6 in clot-stimulated smooth muscle cells [32]. In addition, rivaroxaban and dabigatran prevented thrombin generation in primary human umbilical endothelial cells, leading to downregulation of thrombin-mediated pro-inflammatory cytokine expression to the same extent as protease-activated receptors 1 (PAR-1) antagonist [33]. Apixaban, in turn, was reported to exert an anti-inflammatory effect by reducing the production of free radicals in an in vitro ischemic stress model [34]. In this study, the concentration of IL-6 and pentraxin 3 (PTX3) in the presence of apixaban decreased to the same extent as in the presence of antiplatelet drugs. Since PTX3 is related to vascular inflammation [35], it seems that apixaban may show anti-inflammatory effects comparable to antiplatelet agents [36,37]. Of note, in another study, apixaban and dabigatran were shown to inhibit platelet aggregation following agonist stimulation in vitro, indicating that these DOACs may have not only an anti-inflammatory, but also an antiplatelet effect [38].

The anti-inflammatory effects of DOACs were also shown in clinical studies. In a group of 93 patients with non-valvular AF, treatment with rivaroxaban for 24 weeks was associated with an increase in the concentration of anti-inflammatory thrombomodulin and a trend towards a reduction in pro-inflammatory matrix metalloproteinase 9 as compared to warfarin [39]. In 26 patients with acute ischemic stroke, both dabigatran and apixaban had anti-inflammatory effects by decreasing the stroke-induced inflammatory response as reflected by reduced concentrations of IL-6 and high sensitivity C-reactive protein after 1 week of treatment [40].

### 3.2. Vascular Protection by DOACs

Endothelial injury is a part of the Virchow triad of thrombosis. As mentioned before, DOACs dampen inflammation, thereby protecting vascular endothelial cells. In addition, DOACs were shown to induce vasorelaxation due to increased endothelial nitric oxide synthase (eNOS) activity [41]. Since NO release decreases platelet activation and aggregation, the DOAC-mediated increase in NO release from endothelial cells may also contribute to the effectiveness of DOACs in VTE and stroke prophylaxis [41]. Finally, it was shown that apixaban enhances vasodilation [42]. Although no direct effect of apixaban on endothelial-mediated NO production was observed, vasodilation was mediated through PAR-2 by inhibiting their desensitization [42]. The pleiotropic effects of DOACs are presented in Figure 3.

### 3.3. Efficacy and Safety of DOACs in Stage 3 and 4 CKD

In a recent meta-analysis, which included five hallmark randomized controlled trials comparing DOACs against VKA (ARISTOTLE, ENGAGE AF-TIMI 48, RELY, ROCKET AF, J-ROCKET AF), the subgroup analysis of patients with CKD was performed. This analysis included 12,155 patients with stage 3 CKD (CrCl 30 to 50 mL/min) and 390 patients with stage 4 CKD (CrCl 15 to 30 mL/min). It was observed that DOACs may slightly reduce the rate of all strokes and systemic embolism in comparison to warfarin, both in stage 3 CKD (relative risk (RR) 0.82, 95% confidence interval (CI) 0.66 to 1.02) and in stage 4 CKD (RR 0.68, 95% CI 0.23 to 2.00) [43]. In addition, there was a trend towards decreased all-cause mortality on DOACs in comparison to warfarin in the total population of patients with G3 and G4 CKD (0.91, 95% CI 0.78 to 1.05) [43]. Regarding safety, there was a trend towards a reduced rate of major bleeding events on DOACs in comparison with warfarin in stage 3 CKD (RR 0.80, 95% CI 0.62 to 1.03) [43]. In turn, DOACs reduced a number of major bleeding events in comparison to warfarin in stage 4 CKD (RR 0.30, 95% CI 0.11 to 0.80). Further, DOACs seemed to reduce the rate of intracranial hemorrhage in comparison to warfarin in the total population of CKD patients (RR 0.43, 95% CI 0.27 to 0.69) [43]. However, they led to slightly more gastrointestinal bleeding events (RR 1.40, 95% CI 0.97 to 2.01) [43]. Based on these results, DOACs seem to be as efficient as warfarin to prevent stroke and systemic embolism, without increasing or even decreasing the risk of major bleeding events among AF patients with CKD.

In a recent study, which enrolled 269 patients with atrial fibrillation and advanced chronic kidney disease (defined as CrCl 25 to 30 mL/min), the safety of apixaban versus warfarin was compared. Apixaban caused less major bleeding (hazard ratio, 0.34 (95% CI, 0.14–0.80)) and major or clinically relevant nonmajor bleeding (hazard ratio, 0.35 (95% CI, 0.17–0.72)) compared with warfarin [44]. Patients treated with apixaban demonstrated a trend toward lower rates of major bleeding when compared with those with CrCl > 30 mL/min (P interaction = 0.08) and major or clinically relevant nonmajor bleeding (P interaction = 0.05) [44].

These results encourage the prescription of DOAC in AF patients with stage 3 and 4 CKD.

### 3.4. Efficacy and Safety of DOACs in ESRD

Despite the lack of guidelines on the use of DOACs in patients with ESRD and a lack of prospective trials to evaluate their efficacy and safety, there are some studies reporting the off-label use of DOACs in this population [45].

One study found an increase in prescribing both dabigatran and rivaroxaban in patients undergoing HD shortly after their approval for use in the US, despite the exclusion of patients with ESRD from landmark studies and lack of evidence regarding DOAC safety in this population [46]. This study found an increase in the risk of hospitalization or death due to bleeding with dabigatran (RR, 1.48; 95% CI 1.21–1.81; *p* = 0.0001) and rivaroxaban (RR, 1.38; 95% CI 1.03–1.83; *p* = 0.04) compared to warfarin in the HD population, suggesting that these two drugs are not entirely safe in HD patients. The limitation of this study is that no patient in this study received apixaban or edoxaban. Of note, another study that included 91 patients with AF on chronic HD found contradictory results. Seventy-six of the study participants were initially treated with warfarin, and 15 were initially treated with DOACs (12 with apixaban and 3 with dabigatran). In the warfarin group and dabigatran group, most patients eventually switched to apixaban. Nearly 30% of all patients experienced a bleeding event, and more bleeding events occurred in patients on warfarin compared to those who received initially warfarin and switched to DOACs (mostly apixaban) or who were treated with DOAC from the beginning (*p* = 0.022) [45]. These results indicate that especially apixaban might not increase the risk of bleedings in patients on HD. However, this conclusion is limited by a small sample size included in this study.

In a recent study of 132 patients on HD with AF who were randomized to three groups: (i) a VKA with a target INR of 2–3, (ii) a dose of rivaroxaban 10 mg daily or (iii) rivaroxaban and vitamin K_2_ group. Each therapeutic option was prescribed for 18 months. The primary efficacy endpoint was a sum of fatal and nonfatal cardiovascular events. The primary endpoint occurred at a rate of 63.8 per 100 person-years in the VKA group, 26.2 per 100 person-years in the rivaroxaban group and 21.4 per 100 person-years in the rivaroxaban and vitamin K_2_ group. The estimated competing risk-adjusted hazard ratio for the primary endpoint was 0.41 (95% CI, 0.25 to 0.68; *p* = 0.0006) in the rivaroxaban group and 0.34 (95% CI, 0.19 to 0.61; *p* = 0.0003) in the rivaroxaban and vitamin K_2_ group compared with the VKA group. The hazard ratio for life-threatening and major bleeding compared with the VKA group was 0.39 (95% CI, 0.17 to 0.90; *p* = 0.03) in the rivaroxaban group, 0.48 (95% CI, 0.22 to 1.08; *p* = 0.08) in the rivaroxaban and vitamin K_2_ group and 0.44 (95% CI, 0.23 to 0.85; *p* = 0.02) in the pooled rivaroxaban groups. Results of this study suggest that a reduced dose of rivaroxaban decreased the outcome of fatal and nonfatal cardiovascular events and major bleeding complications compared with VKA in patients on hemodialysis with AF.

In another study, standard and low dose apixaban was examined compared to warfarin in 25,523 patients with AF on HD (2351 patients on apixaban and 23,172 patients on warfarin) [47]. Patients treated with apixaban 5 mg twice daily had a lower risk of stroke or systemic embolism, major bleeding and death compared to warfarin. In turn, a reduced dose of apixaban 2.5 mg twice daily was associated with a lower risk of major bleeding but no difference in stroke or systemic embolism or death. In addition, the authors observed only a modest increase in apixaban exposure off HD and the limited removal of apixaban during HD. These results indicate that apixaban seems to be safe in patients with ESRD and that it might be used in patients with ESRD on HD without dose reduction since a reduced dose is associated with worse stroke prevention [48]. However, in patients with nonvalvular AF and at least one of the following characteristics: age >80 years and body weight <60 kg, it is recommended to reduce the dose of apixaban from 5 mg twice daily to 2.5 mg twice daily, following the recommendations from non-CKD patients [49].

In the case of rivaroxaban, it is recommended to use a reduced dose (15 mg once daily) in patients with a creatinine clearance of <30 mL/min. A study in 16 patients showed that the deterioration of renal filtration function from stage 3 CKD to ESRD does not have a significant impact on rivaroxaban pharmacokinetics and pharmacodynamics. Thus, rivaroxaban 15 mg once daily results in comparable drug exposure for patients with stage 3, stage 4 or ESRD [48]. Another small, prospective study that evaluated pharmacokinetic and pharmacodynamic parameters of apixaban and rivaroxaban found that administration of apixaban at the standard dose (5 mg twice daily) or rivaroxaban in a reduced dose (15 mg once daily) resulted in similar concentrations of these drugs in healthy volunteers and in patients receiving a 3–4-h HD. The concentrations were also comparable to those observed in the landmark trials in patients with AF [45]. Similar results were observed for edoxaban 15 mg once daily. For dabigatran, there were no studies that evaluated the adjustment in patients with advanced CKD [50].

These results were reflected by a revised product labeling of rivaroxaban (August 2016) and apixaban (July 2016) to include the possibility of use for patients treated with intermittent HD.

### 3.5. DOACs Limitations

Extreme body weight may lead to changes in clearance of the medications and may lead to adverse outcomes. Fixed drug doses may lead to decreased drug exposures in obese patients and increased drug exposures in underweight patients based on drug pharmacokinetic changes [51]. The International Society on Thrombosis and Haemostasis conducted an analysis, whose results suggests that DOACs are safe in patients ≤ 120 kg (body mass index ≤ 40 kg/m^2^) at standard doses and are not recommended in patients > 120 kg (body mass index > 40 kg/m^2^) [52]. Based on the data from conducted studies, in the obese population (patients > 120 kg and/or >40 kg/m^2^), it is recommended to avoid the use of dabigatran and edoxaban [51]. Rivaroxaban and apixaban may be used with caution [51].

Findings showed that DOACs were not only more effective, but safer than warfarin for patients, even with extremely low body weight (<50 kg) [53].

Patients with AF may still have an ischemic stroke despite taking DOACs [54]. In secondary prevention of these incidents, left atrial appendage closure is processed.

Advantages and disadvantages of DOACs vs VKA are presented in Table 1. 

## 4. Discussion

Since both CKD and AF are associated with aging, their combined prevalence is increasing. This leads to an increasing number of patients with CKD who require anticoagulation. For many years, the only available group of oral anticoagulants were VKAs. However, a new era of oral anticoagulation began in 2008 with dabigatran and rivaroxaban registration, and now DOACs are replacing VKAs due to a comparable efficacy and better safety profile. The guidelines of the European Society of Cardiology (ESC) and Canadian Cardiovascular Society recommend DOACs over warfarin in patients with mild to moderate CKD. ESC guidelines suggest the preferential use of DOACs in patients with severe CKD but do not provide information regarding patients with ESRD [55]. Both guidelines highlight the difficulty in providing recommendations in the absence of randomized controlled trials. However, based on the available evidence from the meta-analysis, DOACs seem to be non-inferior to VKAs in terms of efficacy and might be superior in terms of safety in patients with stage 3 and 4 CKD. Although these results should be interpreted with caution, they give hope regarding the potential use of DOACs in patients with stage 3–4 CKD.

Although registration trials comparing DOACs with warfarin excluded patients with stage 5 CKD, reports on the off-label use of this group of drugs in ESRD patients provided preliminary data regarding their efficacy and safety. Patients with stage 5 CKD are prescribed warfarin most often (89%), followed by apixaban (9–10%), rivaroxaban (0.8–1%) and dabigatran (0.3–1%) [47,56]. Based on these preliminary results, the updated American guidelines on AF from 2019 provided a class IIb recommendation for the use of either apixaban or warfarin in patients with AF, CHA2DS2-VASc score of at least 2 and stage 5 CKD or those on HD [57]. On the contrary, dabigatran, rivaroxaban or edoxaban should be avoided by this population due to the lack of evidence demonstrating the favorable risk–benefit ratio in patients with ESRD or on HD [57].

In trials that have assessed the effect of revascularization in patients with stable coronary disease, patients with ESRD were routinely excluded. However, in a recent study among patients suffering from stable coronary disease, ESRD and moderate or severe ischemia, it was proven that an initial invasive strategy, as compared with an initial conservative strategy, did not reduce the risk of death or nonfatal myocardial infarction [58]. The results of this study may initiate a change in the treatment strategy in patients with stable coronary artery disease. Conservative treatment, including pharmacological treatment with anticoagulants, may gain in importance.

Since a small number of patients in studies assessing the safety and efficacy of DOACs in ESRD are limiting the results, there is an urgent need for randomized controlled trials to clarify whether DOACs are as efficient and safe in patients with ESRD as in the population with normal kidney function. Some trials comparing the efficacy and safety of apixaban versus vitamin-K antagonists for stroke prevention in patients with AF and ESRD are currently in progress. These studies include the SAFE-HD trial (Strategies for the Management of Atrial Fibrillation in Patients Receiving Hemodialysis; ClinicalTrials.gov (accessed on 2 October 2021) identifier NCT03987711) and the AXADIA trial (Compare Apixaban and Vitamin-K Antagonists in Patients with Atrial Fibrillation and End-Stage Kidney Disease; ClinicalTrials.gov (accessed on 2 October 2021) identifier NCT02933697). Both studies are recruiting patients now.

## 5. Conclusions

Data about safety and efficacy of DOACs in CKD are limited, but there are a few studies in progress that may provide evidence of either superiority or inferiority of DOACs over VKAs. At present, DOACs are preferred over warfarin in patients with mild to moderate CKD and might be considered in patients with advanced CKD. Whereas dabigatran, rivaroxaban and edoxaban should be used in a reduced dose, apixaban may be used in a standard dose. In stage 4, apixaban, rivaroxaban and edoxaban might be used with caution in a reduced dose, whereas dabigatran is contraindicated [18]. European recommendations contraindicate the use of all DOACs in patients with eGFR < 15 mL/min or on hemodialysis, whereas the U.S. Food and Drug Administration allows for apixaban use in those cases [57,59]. Hence, DOACs are gradually replacing VKAs in the prevention of thromboembolic events in patients with CKD due to better safety profile and comparable efficacy. However, treating physicians should be aware of the higher risk for bleeding in the CKD population, regardless of the choice of anticoagulant, and individualize the therapy according to patient risk factors, including HAS-BLED score, history of bleeding events and concomitant antiplatelet therapy [45].

## Figures and Tables

**Figure 1 ijerph-19-01436-f001:**
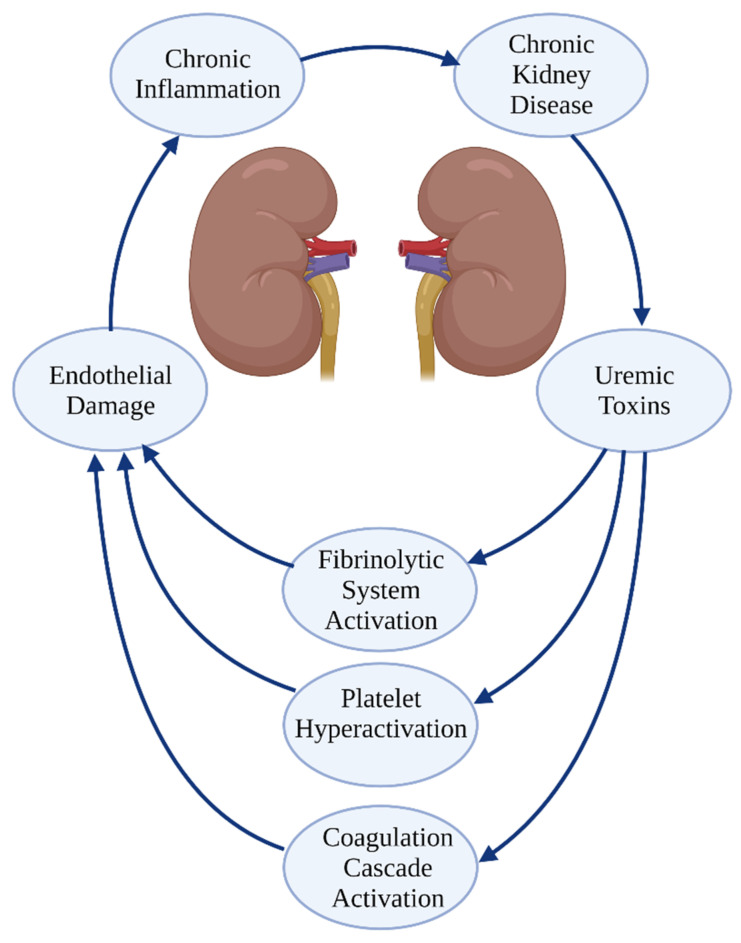
Hemostasis disorders in chronic kidney disease. Created with BioRender.com.

**Figure 2 ijerph-19-01436-f002:**
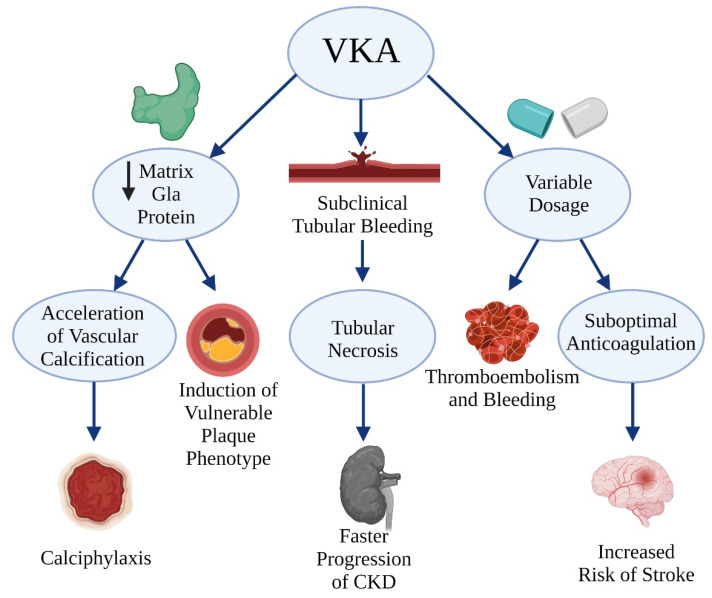
Disadvantages of treatment with vitamin K antagonists in chronic kidney disease (CKD). Created with BioRender.com.

**Figure 3 ijerph-19-01436-f003:**
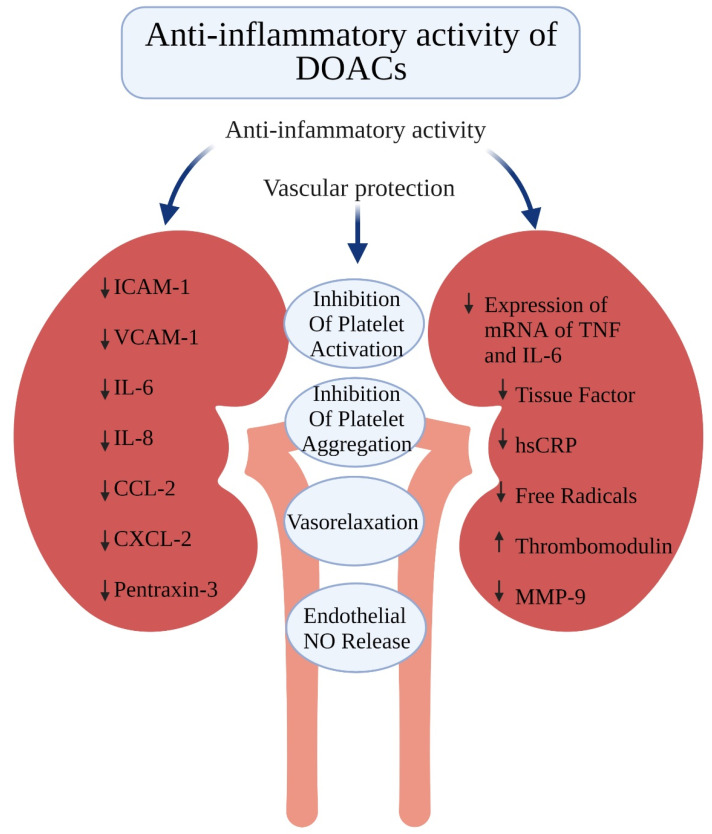
Pleiotropic effects of direct oral anticoagulants. Created with BioRender.com.

**Table 1 ijerph-19-01436-t001:** Summary of DOAC advantages and disadvantages versus VKAs. DOAC—Direct Oral AntiCoagulants, VKAs—Vitamin K Antagonists.

Agent	Advantages	Disadvantages
DOACs	Anti-inflammatory effect Vascular protection Predictable effect Rapid onset and offset No dietary precautions Wide therapeutic window	High cost Decreased exposure in obese patients
VKAs	Wide clinical experience	Slow onset and offset Unpredictable effect Vascular calcification Multiple drugs and diet interactions Narrow therapeutic window

## Data Availability

Not applicable.

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
