# Peer review of "Safety and Efficacy of DOACs in Patients with Advanced and End-Stage Renal Disease"

_ijerph, 2022, doi:10.3390/ijerph19031436_

Round 1

Reviewer 1 Report

I have read the review article conducted by Rogula et al. with great interest. The article comprehensively reviews the safety and efficacy of direct oral anticoagulants in patients with advanced and end-stage renal disease. The review is excellent and brilliant, describing the mechanism of hemostasis disorders in chronic kidney disease, disadvantages of treatment with vitamin K antagonists in chronic kidney disease, and pleiotropic effects of direct oral anticoagulants in detail. The attached figures are readable and clear, that will be helpful for the readers.

I only have one minor comment:

Line 77: Reference 18, the ESC Guidelines for the diagnosis and management of atrial fibrillation developed in collaboration with the European Association of Cardio-Thoracic Surgery (EACTS) have updated to 2020 version. Some changes of recommendation in CKD have declared through the update version, especially in patients with stage 4 CKD, please revised accordingly.

Reviewer 2 Report

The article is a review that evaluates the safety and efficacy of direct oral anticoagulants (DOACs) in patients with advanced and end-stage chronic kidney disease (CKD/ESRD) in an attempt to replace Vitamin K antagonists (VKAs).

The literature lacks research clinical trials (RCT) comparing these therapies because patients with advanced CKD are excluded in most RCTs. The review is comprehensive, concise, and highlights the therapeutic advantages and difficulties of using these agents in this special group of patients.

In the text that evaluates the VKAs, it would be interesting to mention the role of fibroblast growth factor 23 (FGF-23) and its protagonist role in influencing and aggravating CKD early on due to the increase in its circulating levels already in stages 2 of CKD.

The use of VKAs accelerates the evolution of calcification and therefore CKD, in addition to the fact that FGF-23 can increase the incidence of atrial fibrillation. See: Wolf M et al. Kidney Int 2012, 82(7): 737-747 and Mathew JS et al. Circulation 2014, 130 (4): 298-307).  

Subclinical renal bleeding, more common with the use of VKAs, is known as warfarin related nephropathy and may occur in the absence of evidence of hemorrhage, initially described by Brodsky SV et al. Kidney Int 2011, 80: 181-189.

The review highlighted the pleiotrophic effect of DOACs and enriched the preference of this class of drugs in CKD. The final part of the review assesses the safety and efficacy of DOACs in patients with CKD/EARD, limitations and ends with a general discussion and conclusions consistent with the purpose of the review.

We suggest approval of the article.

Reviewer 3 Report

I read with interest authors' work, that is a review about the available data on the efficacy and safety of DOACs in patients with CKD. The paper provides recommendations regarding the choice of the optimal drug and dosage, depending on the CKD stage. The authors had made a hard work to summarize some of the immense literature on CKD. It seems that there is no major concerns found in the analyzed sections. the study lacks a summary on the benefits that DOACs would offer as compared with the competitors (consider adding tables). English might be improved, even if it is acceptable in its current form
